# CFDLLMBench: A Benchmark Suite for Evaluating Large Language Models in Computational Fluid Dynamics

## Abstract

Large Language Models (LLMs) have demonstrated strong performance across general NLP tasks, but their utility in automating numerical experiments of complex physical system—a critical and labor-intensive component—remains underexplored. As the major workhorse of computational science over the past decades, Computational Fluid Dynamics (CFD) offers a uniquely challenging testbed for evaluating the scientific capabilities of LLMs. We introduce *CFDLLMBench*, a benchmark suite comprising three complementary components—*CFDQuery*, *CFD-CodeBench*, and *FoamBench*—designed to holistically evaluate LLM performance across three key competencies: graduate-level CFD knowledge, numerical and physical reasoning of CFD, and context-dependent implementation of CFD workflows. Grounded in real-world CFD practices, our benchmark combines a detailed task taxonomy with a rigorous evaluation framework to deliver reproducible results and quantify LLM performance across code executability, solution accuracy, and numerical convergence behavior. *CFDLLMBench* establishes a solid foundation for the development and evaluation of LLM-driven automation of numerical experiments for complex physical systems.

## 1 Introduction

Recent advances in large language models (LLMs) have shown remarkable performance across general natural language processing tasks [19, 1]. However, their potential as scientific assistants—specifically, their ability to automate numerical simulation workflows—remains largely underexplored [10, 25]. Computational Fluid Dynamics (CFD) is critical in domains such as urban physics [7, 6], aerospace [46], climate [42], and aerial [43] and underwater robotics [28], and has labor-intensive workflows for computationally expensive numerical simulations of fluid dynamics. CFD workflows involve multiple steps, such as mesh generation, setup of boundary and initial conditions, and solver configuration. Such scientific workflows require an understanding of highly specialized knowledge [51], numerical and physical reasoning [50], and have context-dependent implementations involving domain-specific tool calling [20].

In this paper, we introduce *CFDLLMBench* (Figure 1), the first LLM benchmark for CFD composed of curated datasets designed to holistically evaluate LLMs' performance across three key competencies:

**Graduate-level CFD knowledge:** Understanding of fluid mechanics and concepts of numerical analysis relevant to CFD.

**Numerical and physical reasoning:** Applying advanced math and physics knowledge to solve difficult problems. For example, selecting a suitable numerical method that solves the governing equation, with the appropriate boundary conditions and initial conditions.

Submitted to 39th Conference on Neural Information Processing Systems (NeurIPS 2025). Do not distribute.

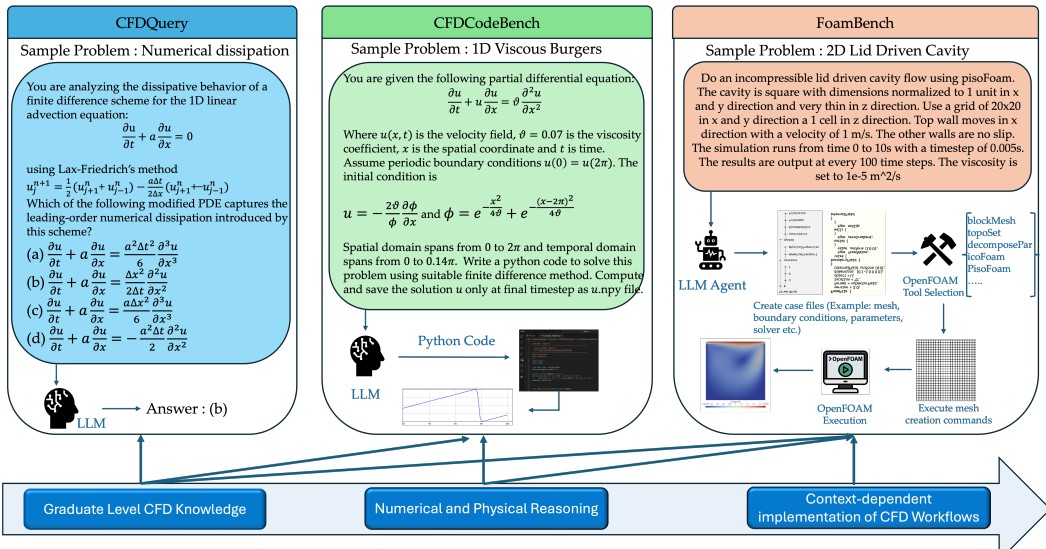

**Figure 1: Overview of CFDLLMBench**: As the first ever LLM benchmark designed to holistically evaluate LLM's capabilities for CFD, it consists of three different tasks and datasets. (1) *CFDQuery*: Graduate-level CFD QA. (2) *CFDCodeBench*: Coding questions about solving common linear/nonlinear PDEs encountered in CFD. (3) *FoamBench*: Configuring OpenFOAM case files for simulating realistic engineering scenarios such as incompressible flow over obstacles, supersonic flow with shockwaves, Rayleigh-Benard convection, etc.

**Context-dependent implementation of CFD workflows:** Selecting and configuring CFD preprocessing and numerical solver settings according to physical context.

The *CFDLLMBench* benchmark suite evaluates these competencies using three benchmark tasks: **1) CFDQuery:** 90 multiple-choice questions curated from graduate-level CFD lecture notes that assess LLM's ability in the conceptual understanding of CFD knowledge. **2) CFDCodeBench:** 24 CFD programming tasks designed to assess an LLM's ability to generate correct simulation code from descriptions of physical problems. **3) FoamBench:** 110 basic and 16 advanced numerical simulation tasks, drawn from practical engineering problems, designed to assess the LLM's ability to implement OpenFOAM [53] workflows. OpenFOAM projects typically have 6-7 configuration files, totaling ~300-600 lines of code per case.

Although strong performance on *CFDQuery* indicates excellent recall of relevant CFD knowledge, success in solving *CFDCodeBench* and *FoamBench* would suggest that LLM possesses reasoning and workflow implementation capabilities near the proficiency of a competent CFD assistant. To support a holistic evaluation of these diverse benchmark tasks, we equip each benchmark task with one or more tailored metrics, which are developed in collaboration with CFD experts.

We use *CFDLLMBench* to evaluate both state-of-the-art proprietary and open-source LLMs. Despite relatively strong performance on *CFDQuery*, the results highlight the challenge of the latter two tasks (see Figure 2): the best performing model achieves only **14%** on *CFDCodeBench* and **34%** on *FoamBench*. In the more complex *FoamBench Advanced* split, generally, performance is poor, e.g., Gemini 2.5 Flash drops to **0%**. In *FoamBench*, all models show major improvement when deployed in a multi-agent framework, as opposed to zero-shot prompting (near 0 performance).

The remainder of the paper is organized as follows. Section 2 describes related work. Section 3 presents our holistic CFD benchmark. Section 4 summarizes our experimental setup and results, which are discussed in Section 5. Section 6 has limitations and Section 7 concludes the paper.

## 2    Related Work

**LLMs for science & engineering**    LLMs are becoming increasingly proficient at knowledge-intensive tasks in general science [48, 5, 33, 45] and engineering [21], aided by dedicated pretraining on scientific corpora. The development of language agents with tool-use [40, 9, 10, 36] further

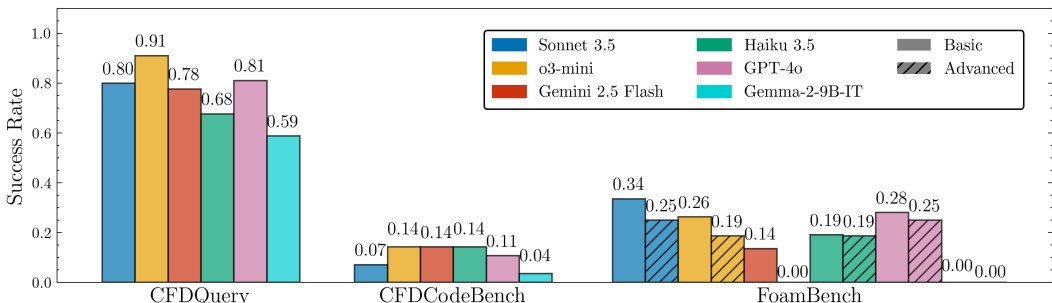

**Figure 2:** Success Rate comparison of different models across the three tasks. Success Rate is the fraction of cases in the benchmark that produce physically accurate results (higher is better). The detailed definition of Success Rate for each benchmark task can be found in section 3.3. The results for *FoamBench* are produced using the *Foam-Agent* framework with RAG, Reviewer, and Sonnet 3.5. There is a steep drop in performance from graduate-level knowledge (*CFDQuery*) to practical simulation workflow automation *FoamBench*.

enhances LLMs' capabilities, enabling them to integrate with complex scientific and engineering software [15]. Recent work explores the use of LLMs to generate input files in domain-specific languages for quantum chemistry [20] and building energy [23] simulators, tasks which demand substantial time from a researcher to master. LLMs are also accelerating workflow automation in computational physics. MyCrunchGPT [25] demonstrates the use of automated scientific machine learning workflows to optimize airfoils in aerodynamics. MetaOpenFOAM [13], OpenFOAMGPT [39], and Foam-Agent [54] exemplify this trend by automatically configuring and conducting complex CFD simulations based on human requests. These examples highlight the critical need for effective workflow automation benchmarking.

**LLM benchmarks for science & engineering** Recent interest in the use of LLMs in science and engineering has led to benchmarks measuring specific advanced LLM capabilities such as graduate-level scientific problem solving [41, 52, 18, 55] and long-context reasoning [29, 16]. Our benchmark aims at practicality, providing a holistic evaluation that includes a real-world numerical simulation workflow automation task. Other related workflow benchmarks focus on paper reproduction [47, 8, 44] or data analysis workflows [14, 34, 35]. Paper reproduction, data analysis, and simulation automation (ours) are all critical workflows in the scientific discovery life cycle. Differently, our benchmark uniquely evaluates numerical and physical reasoning, an underexplored capability in LLMs. Thus, these benchmarks assess distinct yet complementary capabilities for scientific workflow automation. The most closely related benchmark is FEABench [31], which evaluates the ability of LLMs as agents for solving PDEs using COMSOL, a commercial finite element analysis software that requires a license of several thousand dollars per year. In contrast, our work is a comprehensive benchmark that consists of domain-specific knowledge, reasoning, and OpenFOAM [22]workflow automation, one of the most widely used open-source numerical simulation software.

**LLM benchmarks for code generation** Code generation benchmarks such as MBPP [3], HumanEval [12], DS-1000 [27], and SWE-Bench [24] evaluate general coding yet lack the complexity of scientific and engineering tasks. These require understanding advanced concepts and implementing sophisticated algorithms that involve specialized libraries. SciCode [50] is a related scientific coding benchmark, but their CFD examples-1D heat transfer and 1D Burgers equation-are far from enough to represent the algorithmic, physical, and geometrical complexity in CFD. There is a clear need for a comprehensive code generation benchmark that meets the scientific standards for CFD.

## 3 CFDLLMBench: a benchmark suite for evaluating LLMs in CFD

We present *CFDLLMBench*, which holistically assesses three capabilities of LLMs necessary to perform CFD-related tasks (Figure 1). We begin with *CFDQuery* which evaluates graduate-level conceptual understanding, after which the benchmark progresses to the application of this knowledge through *CFDCodeBench*, where LLMs must use numerical and physical reasoning over a description of a physical problem to correctly generate CFD code in Python. Finally, the most practical and challenging benchmark task is *FoamBench*, where LLMs write input files for a CFD software suite

that must correctly pre-process, configure, and execute simulations given physical context expressed in natural language.

**OpenFOAM**    OpenFOAM [53] is an open-source, license-free CFD software suite (a collection of software for fluid-flow simulation that covers meshing, solving, and post-processing) widely used in academia and industry. OpenFOAM projects have a precise file organization and various configuration and source files arranged in a strict folder hierarchy. OpenFOAM's accessibility, extensibility, and rich community resources make it an attractive platform for an LLM benchmark. However, automating OpenFOAM workflows poses significant challenges for language models and agents. Writing code for OpenFOAM requires long-context understanding to track simulation parameters across multiple files, domain-specific tool usage, and accurate implementation of complex physical models. The third benchmark task in our suite, *FoamBench*, uses OpenFOAM as the underlying CFD software suite.

## 3.1    Datasets Overview

**CFDQuery**    This dataset consists of 108 multiple-choice questions pertaining to CFD curated by three domain experts. These questions probe core concepts in fluid mechanics, linear algebra, and numerical methods, with source materials adapted from both web-scraped content and CFD lecture notes. The solution to these problems require the LLMs to have deep knowledge about topics in CFD like linear algebra, numerical methods and fluid dynamics.

**CFDCodeBench**    This dataset consists of 24 CFD problems that require LLMs to generate Python code for their numerical solution. Each problem is described in natural language and specifies the governing Partial Differential Equation (PDE), boundary and initial conditions, the spatial and temporal domain, and the target variable(s) to be computed and saved. The dataset includes both 1D and 2D problems, spanning linear and nonlinear PDEs, representative of those encountered in the CFD domain. Reference solutions are provided either as closed-form analytical expressions or as expert-authored Python implementations. Further details can be found in Appendix A.2. Our 24 coding problems span fluid mechanics, thermal transport, and turbulence, include both 1D and 2D simulation scenarios, extending beyond prior work in terms of complexity, which only evaluates the 1D heat transfer and 1D Burgers' equation [50], in both scope and complexity. Solving these problems requires not only reasoning about the physics but also integrating numerical methods, discretization schemes, and data handling into coherent, executable Python scripts containing 70 lines of code on average per problem.

**FoamBench**    This task requires LLMs to generate all input files for an OpenFOAM simulation using the proper project folder structure and for the simulation to execute correctly, producing a physically accurate result with respect to a reference project. It consists of 126 OpenFOAM cases spread over more than 15 distinct geometric and physics scenarios. This dataset is further divided into two. *(1) FoamBench Basic*: This consists of 110 OpenFoam cases obtained from 11 tutorial cases [53]. We create variations within them by altering the boundary conditions and the parametric values on a case-specific basis (more details can be found in Appendix A.3.1). *(2) FoamBench Advanced*: This consists of 16 challenging OpenFOAM cases, which are not similar to the tutorials and are hand-crafted by CFD experts. Unlike *Basic*, the *Advanced* split tasks LLMs with choosing a proper turbulence model, creating a new geometry, and creating an appropriate mesh, based on the natural language input, without potentially relying on a tutorial project for guidance. For example, in the *Advanced* flow over double square case, the prompt specifies two square obstacles with details of their location and size. The LLM must correctly understand this prompt, then use appropriate one or more meshing tools from the OpenFOAM suite (e.g. blockMesh) to generate a valid computational mesh. Such cases bring us closer to real-world scenarios, where engineers analyze flow over complex geometries based on design specifications. Further details of the cases are provided in Appendix A.3.2.

For each case in *FoamBench*, the prompt (Appendix A.3.1) is designed to be concise and sufficient. The prompt contains (1) a clear description of the problem (e.g., flow over a cylinder), physical scenario (compressible or incompressible), geometry including computational domain and obstacle locations (with retrieval mechanisms handling complex geometries) and specifies the exact Open-FOAM solver for consistency; (2) the boundary conditions, relevant parameters (viscosity, Prandtl number), turbulence models (e.g., $k - \epsilon$, SA, LES), and specifies the timestep and solution-saving intervals for comparison against reference solutions.

## 3.2 Dataset Creation

In this section, we describe the dataset curation process. Due to the complex and technical nature of our benchmark, we relied on human experts at several stages during the creation of *CFDLLMBench*, involving them in both curation of data from existing sources, as well as authoring new content for the benchmark. A complete description of our process is presented in Appendix A.

**Expert contributors**   For all three datasets, human experts curated or authored the initial set of problems. Our team of experts included six domain experts with advanced degrees and professional experience in the field of CFD, including two doctoral students, one Master's student, one undergraduate student, one post-doctoral researcher, and one university professor, with the latter two reviewing the work of the other four at each step. Despite being experts in CFD, they were still provided an orientation ahead of the curation process. For *CFDQuery*, the human experts created the multiple choice problems, and for *CFDCodeBench*, the human experts authored descriptions for the advanced problems by reviewing the source code. For *FoamBench*, the experts curated the dataset by varying parameters and boundary conditions for the tutorial problems, designing novel geometries for the non-tutorial cases, and authoring corresponding prompts based on the case files to guide LLMs in generating valid simulation setups. While the nature of the human work did not warrant an IRB review, we nevertheless followed all ethical norms and standards of the host academic institute when performing the human tasks for this dataset. All human experts involved are individuals involved in the project and well-compensated for their time.

**Data sources**   For this benchmark, we ensured that we only used highly vetted data sources. The *CFDQuery* dataset was created exclusively for this benchmark, but the reference sources include university-level CFD lecture notes and vetted online sources. The problems in *CFDCodeBench* were curated from publicly available GitHub repositories and established numerical solver packages, including *CFD Python: the 12 Steps to Navier-Stokes Equations* repository [4] and *ENGR 491 - Computational Fluid Dynamics*, while more challenging scenarios were curated from the Dedalus Project [11]. For *FoamBench*, we curated the dataset based on the 11 OpenFOAM tutorials [53].

**Quality assessment**   Since the solutions to our problems include objective, scientific answers, we did not perform traditional measures of human agreement. Rather, we went through an iterative process of review and revision of human work by independent experts to ensure the quality of the work. This review included both human-curated and human-authored portions of the benchmark.

## 3.3 Evaluation Metrics

Here we define expert-informed metrics used to assess performance on *CFDLLMBench*.

**CFDQuery**   We evaluate multiple choice accuracy using a single standard accuracy metric, *Success Rate*, defined as ratio of the number of correctly answered questions to the total number of questions.

**CFDCodeBench**   We evaluate an LLM's ability to generate executable and physically accurate python code for the numerical solution of a given CFD problem using four metrics. The holistic metric we use has three components: code executability, relative numerical error, and numerical convergence. We aggregate these three into a single score, which we call the *Success Rate*. **1) Executability** ($M_{\text{exec}}$): This is a binary metric which takes on a value of 1 if the LLM generated python code executes successfully and 0 is it is a failure. This metric is akin to the common pass@1 metric [12]. **2) Relative Error** ($M_{\text{NMSE}}$): We compare the LLM generated solution to the reference solution at the *final time of the prescribed simulation interval*. A normalized mean squared error percentage is calculated and a score is assigned based on the value of the NMSE percentage given by

$$\text{NMSE\%} = \frac{\sum_{i=1}^{N}(y_i - \hat{y}_i)^2}{\sum_{i=1}^{N} y_i^2} \times 100, \quad M_{\text{NMSE}} = \begin{cases} 1, & \text{NMSE} \leq 10\%\,, \\ 0.5, & 10\% < \text{NMSE} \leq 30\%\,, \\ 0, & \text{NMSE} > 30\%\,. \end{cases} \quad (1)$$

An $M_{\text{NMSE}}$ of 0 means the solution is not physically accurate while a score of 0.5 is considered partial success. A score of 1 means the solution is acceptably accurate. **3) Numerical convergence** ($M_{\text{conv}}$): To evaluate the numerical convergence of the solution generated by the LLM, we refine both

the spatial and temporal discretization and assess the corresponding change in relative error. If the error decreases with mesh and time-step refinement, the solution is deemed convergent and awarded a score of 1; otherwise, it receives a score of 0. Unlike conventional LLM code generation benchmarks, we cannot rely on code similarity with respect to a reference solution, as numerical simulation code can vary significantly in implementation while yielding identical or equivalent solutions. **4) Success Rate**: We also define a stringent criterion to assess successful runs by looking at the fraction of problems where *all three* metrics achieve a score of 1. Specifically, defined for each problem $i$:

$$M_{\text{success}}^{(i)} = \begin{cases} 1, & M_{\text{exec}}^{(i)} = 1 \ \wedge \ M_{\text{NMSE}}^{(i)} = 1 \ \wedge \ M_{\text{conv}}^{(i)} = 1, \\ 0, & \text{otherwise}, \end{cases} \quad \text{Success Rate} = \frac{1}{K} \sum_{i=1}^{K} M_{\text{success}}^{(i)}, \tag{2}$$

where $K$ is the total number of problems. This provides us with a stringent measure of the percentage of problems within the benchmark where the model was able to produce an executable, physically accurate, and convergent solution.

**FoamBench**   This task requires an LLM to create the required OpenFOAM input files, save them in appropriate directories, and call different solvers and tools within OpenFOAM to run a physically accurate simulation, all based on a natural language prompt. Prior work [13] focuses only on the ability of LLMs to generate files that produces a successful execution of OpenFOAM. Though executability is important, it does not capture the physical accuracy of the generated solution and thus fails to provide insights into whether the solution satisfies the user requirements. Text similarity metrics are widely used in comparing LLM-generated text to human text. For code generation, this is a useful metric for giving us an idea of how complete the files generated by LLMs are in comparison to the reference files, but again fails to provide the complete picture.

To tackle these challenges, we use four metrics to evaluate the LLM generated code, capturing code quality and physical accuracy of the solution, plus a holistic statistic, Success Rate. The details are as follows. **1) Executability** ($M_{\text{exec}}$): Similar to *CFDCodeBench*, we assign a value of 1 for successful execution of OpenFOAM using LLM generated case files and 0 otherwise. **2) Folder and File Structure** ($M_{\text{struct}}$): Generating the correct files and placing them in their respective folders is critical to the successful and accurate execution of the simulation workflow. The absence or misplacement of files can lead to failed execution of the case and/or inaccuracy of the generated output. Here, we use the ROUGE similarity metric [32] to compare the reference folder structure of the OpenFOAM cases with the LLM generated folder structure and provide a score between 0 and 1. **3) File Similarity** ($M_{\text{file}}$): This metric compares the content of the generated files with the reference OpenFOAM files using the ROUGE metric. **4) Relative Error** ($M_{\text{NMSE}}$): We use the same approach as *CFDCodeBench* Equation (1), comparing the LLM generated solution to a reference solution at the final time of the prescribed simulation window. **5) Success Rate**: We define Success Rate as the fraction of cases where just $M_{\text{exec}}$ and $M_{\text{NMSE}}$ achieves a score of 1.

## 3.4   Licensing, Accessibility, and Usability

All problems in our benchmark were collected from open, publicly available sources or were authored specifically for this benchmark. Accordingly, *CFDLLMBench* is released under the terms of the BSD 3-Clause License, making it free to use, modify, and redistribute, including for commercial purposes, provided that the license conditions are met. Our benchmark pipeline relies exclusively on free and open-source software, ensuring that it is accessible to all users without the need for paid subscriptions. Furthermore, we release not only the dataset (prompts), but also the complete codebase, fully containerized with Docker, to enable reproducibility. This comprehensive release allows future researchers to easily utilize, reproduce, or extend our benchmark with minimal overhead.

## 4   Experiments

In this section, we present results across a wide range of LLMs and agent frameworks that demonstrate the difficulty and realism of our benchmark.

## 4.1 Experimental Setup

For benchmark tasks, we compare the performance of five closed-weight models including Claude Sonnet 3.5 [2], o3-mini [38], Gemini 2.5 Flash [17], Claude Haiku 3.5 [2], and GPT-4o [37], and one open-source model Gemma-2-9B-IT [49]. The temperature parameter is set to 0.0 for the models in evaluation in all experiments, except for o3-mini, which does not allow us to change the default temperature parameters and the value of this parameter is undisclosed. On *CFDQuery* and *CFDCodeBench*, LLMs use a standard zero-shot prompt template that describes the task and the output format. For *FoamBench*, we evaluate LLMs zero-shot, as well as with agentic frameworks (described next). We use OpenFOAM v10 for all experiments.

**Agentic frameworks for FoamBench**    Automating OpenFOAM using LLM is a complicated task, which we find benefits from agentic frameworks. Hence, for *FoamBench*, we not only compare various LLMs, but we also compare two agentic frameworks: *MetaOpenFoam* [13] and *Foam-Agent* [54]. Both of them assign agent roles for Retrieval-Augmented Generation (RAG) [30] , file generation, running, and reviewing (Reviewer). These components enable the system to retrieve files from similar simulations to use as exemplars and to get intermediate feedback for re-attempting file generation if necessary. To assess the individual contributions of these components, we benchmark three configurations: (1) with RAG, with Reviewer; (2) with RAG, without Reviewer (3) without RAG, with Reviewer. The absence of RAG and Reviewer indicates zero-shot LLM prompting-based generation, which is used as a baseline to compare the improvements due to these agent roles.

## 4.2 Results

The Success Rate of different models for the three benchmark tasks is shown in Figure 2. The *FoamBench* results are from the Foam-Agent framework, consisting of RAG and Reviewer, and using Sonnet 3.5, as this configuration yielded the strongest performance in our evaluations. Detailed *FoamBench* results are shown in Table 4. All closed-weight models perform well on *CFDQuery*, while the open sourced model could only answer 60% of the questions correctly. O3-mini performs the best in this task, which is not unexpected as it excels at logical reasoning and structured responses, producing 92% correct answers. On *CFDCodebench* and *FoamBench*, we see a drastic fall in Success Rate dropping to 14% in *CFDCodeBench* and 34% *FoamBench Basic* and 25% in *FoamBench Advanced* for the best performing models. It is interesting to note that Sonnet 3.5 performs the best among other models by some margin in *FoamBench*, which is not seen in the other tasks. However, it costs higher per run on average ($6.56) than, e.g., GPT-4o ($0.42)-see Table 5.

**CFDCodeBench**    Figure 3 illustrates the breakdown of metric scores and Success Rate as defined in Section 3.3 for different models. The accuracy and convergence metrics highlight the importance of holistic evaluation beyond syntactic correctness, which is often lacking in studies.

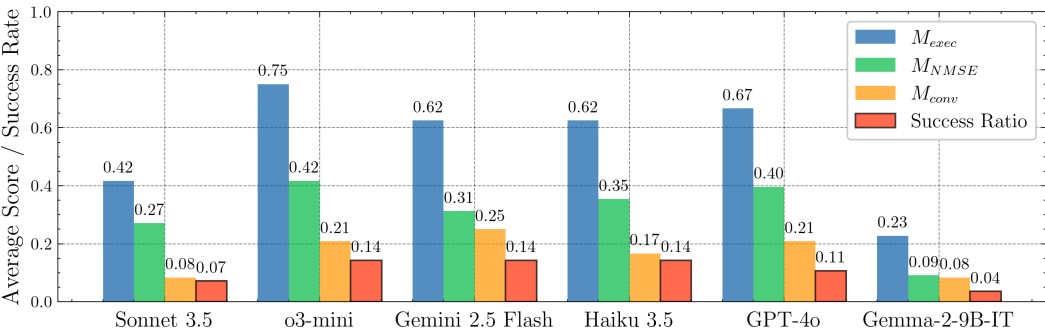

**Figure 3:** Average metric score and Success Rate for *CFDCodeBench*. The Success Rate for even the best performing models are around 14%, suggesting the challenging nature of the problems in this benchmark.

**FoamBench**    Average metric scores and Success Rate of different models using the *Foam-Agent* framework with RAG and Reviewer is shown in Figure 4. Sonnet 3.5 was found to the best performing model for *FoamBench* tasks. The results of non-agentic zero-shot prompting with Sonnet 3.5

is provided in Table 1 to serve as a baseline for improvements due to the RAG and Reviewer roles (Table 2). This table also shows a comprehensive comparison between the two frameworks, MetaOpenFOAM and Foam-Agent, on *FoamBench* Basic and Advanced datasets. Detailed results on the impact of different models, framework and variations are provided in Appendix B.1.

**Table 1:** Zero-shot prompt LLM performance with Sonnet 3.5 (best performing model) on *FoamBench Basic* and *Advanced*.

| Dataset | $M_{\text{exec}}$ | $M_{\text{struct}}$ | $M_{\text{file}}$ | $M_{\text{NMSE}}$ | *Success Rate* |
|---|---|---|---|---|---|
| FoamBench Basic | 0.064 | 0.670 | 0.506 | 0.050 | *0.045* |
| FoamBench Advanced | 0.017 | 0.773 | 0.573 | 0.009 | *0.007* |

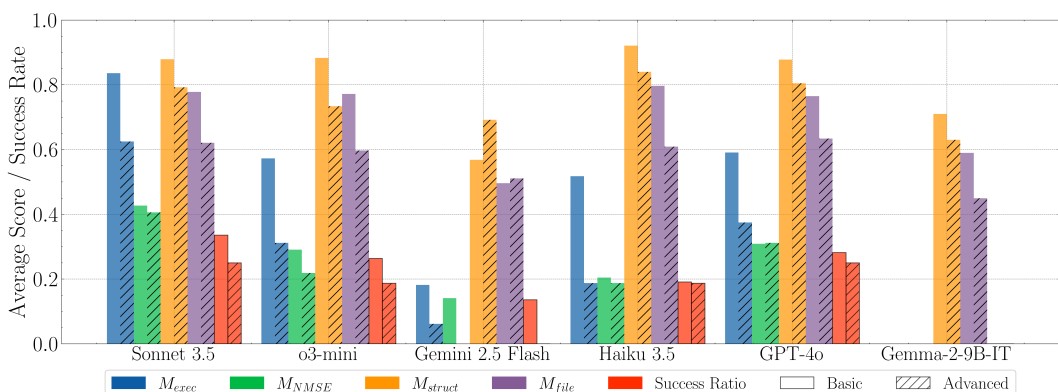

**Figure 4:** Average metric score and Success Rate for different models on *FoamBench* using *Foam-Agent* framework with RAG and reviewer. The Success Rate for even the best performing model (Sonnet 3.5) is 34% in basic dataset and 25% in the advanced dataset.

## 5 Discussion

**Importance of physical and numerical accuracy metrics**    While all models demonstrate strong performance on *CFDQuery*—with Success Rate ranging from 60% (Gemma-2-9B-IT) to 92% (o3-mini), *performance significantly declines on tasks requiring physical and numerical accuracy*. To provide a holistic evaluation of model performance in *CFDCodeBench* and *FoamBench*, we reported multiple metrics and the stricter Success Rate. The latter aggregates success across code executability $M_{\text{exec}}$, numerical convergence $M_{\text{conv}}$, and physical accuracy $M_{\text{NMSE}}$, offering a practical view of model capabilities. From Figure 3, it is evident that most closed-weights models produce executable Python code in over 60% of cases, but these numbers are significantly worse for physical and numerical accuracy. For instance, in *FoamBench Basic*, the best Foam-Agent (Table 2) achieves good coding metrics $M_{\text{exec}} = 0.836$, $M_{\text{struct}} = 0.879$, $M_{\text{file}} = 0.778$, but the Success Rate is only 34% because of low physical accuracy. We see that the LLMs often fail to fully understand the prompts and lack domain-specific reasoning required to correctly apply fundamental CFD concepts—such as flux discretization schemes, appropriate time integration strategies, and consistent boundary treatments. This highlights a critical gap in current models' capabilities when it comes to generating reliable and physically consistent CFD code.

**Zero-shot prompting for OpenFOAM**    Zero-shot prompting produces close to 0% Success Rate even for the best performing model (Sonnet 3.5) as shown in Table 1, highlighting the need for agentic frameworks when it comes to running OpenFOAM. For example, it is difficult for current LLMs to produce all of the required input files in a zero-shot manner. We observe that Sonnet 3.5 and o3-mini (Appendix B.1) have the most successful zero-shot runs.

**Role of RAG and Reviewer**    RAG provides the framework with similar simulation files and the Reviewer allows for a trial and error approach to running OpenFOAM cases, mimicking human troubleshooting. The absence of either decreases the Success Rate by approximately 10% (Table 2), underscoring their critical roles in achieving optimal performance within the proposed framework.

**Table 2:** Component-wise mean scores and Success Rate for Claude Sonnet 3.5 on *FoamBench* Basic and Advanced, comparing MetaOpenFOAM vs. Foam-Agent.

| Dataset | Variation | MetaOpenFOAM | | | | | Foam-Agent | | | | |
|---|---|---|---|---|---|---|---|---|---|---|---|
| | | $M_\text{exec}$ | $M_\text{struct}$ | $M_\text{file}$ | $M_\text{NMSE}$ | *Success Rate* | $M_\text{exec}$ | $M_\text{struct}$ | $M_\text{file}$ | $M_\text{NMSE}$ | *Success Rate* |
| FoamBench Basic | RAG + Reviewer | 0.555 | 0.883 | 0.763 | 0.173 | *0.136* | 0.836 | 0.879 | 0.778 | 0.427 | *0.336* |
| | RAG + No Reviewer | 0.064 | 0.810 | 0.728 | 0.023 | *0.009* | 0.373 | 0.668 | 0.599 | 0.232 | *0.200* |
| | No RAG + Reviewer | 0.400 | 0.747 | 0.522 | 0.195 | *0.145* | 0.473 | 0.862 | 0.647 | 0.291 | *0.245* |
| FoamBench Advanced | RAG + Reviewer | 0.125 | 0.775 | 0.599 | 0.125 | *0.125* | 0.625 | 0.792 | 0.621 | 0.406 | *0.250* |
| | RAG + No Reviewer | 0.000 | 0.743 | 0.594 | 0.000 | *0.000* | 0.188 | 0.771 | 0.609 | 0.156 | *0.125* |
| | No RAG + Reviewer | 0.375 | 0.655 | 0.451 | 0.344 | *0.187* | 0.250 | 0.806 | 0.592 | 0.188 | *0.125* |

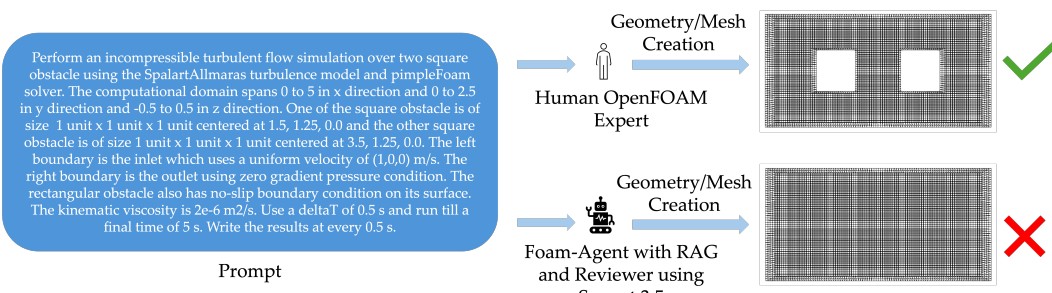

Prompt

**Figure 5:** Comparison of the geometry and mesh generated by the *Foam-Agent* [54] (RAG and Reviewer) with Sonnet 3.5 for the doubleSquare case against human expert.

**Spatial reasoning**    The CFD simulation workflows in *FoamBench* have preprocessing steps where a correct geometry and mesh file must be generated by the LLM. To handle real-world workflows, LLMs should be able to extrapolate to novel geometries. We highlight a particular case from *FoamBench Advanced*, doubleSquare, which is an incompressible flow over two square obstacles. The geometry produced by the *Foam-Agent*, in comparison to the reference geometry, is visualized in Figure 5. The prompt clearly defines the location of the obstacles, but the lack of spatial reasoning capabilities in LLMs appears to produce an incorrect geometry and mesh. We highlight that the ability of LLMs to understand geometry is a major area in need of improvement.

# 6    Limitations

First, one limitation of our work is that we currently do not provide human baselines for benchmark tasks. This is primarily due to the difficulty in determining appropriate human baselines, since the ability of a human to solve these problems depends on their domain knowledge, which is hard to quantify. For example, in a future iteration of the benchmark, we may explore adding a human baseline measuring the time taken for experts to solve these problems. Second, we did not perform an extensive automated prompt tuning for the baselines. Additional prompt engineering for the tasks, as well as automatic design of an agentic framework, may lead to stronger baseline performance.

# 7    Conclusion

In this work, we introduced *CFDLLMBench*, the first benchmark to holistically evaluate graduate-level knowledge, numerical and physical reasoning, and practical simulation capabilities of LLMs for CFD. We accomplish this by structuring the benchmark into three progressively challenging tiers, namely, *CFDQuery*, *CFDCodeBench*, and *FoamBench*. Our results highlight both the promise and the current limitations of LLMs in solving advanced scientific workflow automation problems, which require software expertise such as tool-calling and long-context understanding, as well as accurate physical modeling. We expect that *CFDLLMBench* will serve as a valuable testbed for advancing LLM capabilities in scientific computing, and encourage future work on domain-grounded, execution-based benchmarks across other areas of science and engineering.

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
