# OpenReview forum: "CFDLLMBench: A Benchmark Suite for Evaluating Large Language Models in Computational Fluid Dynamics"
_NeurIPS.cc/2025/Datasets_and_Benchmarks_Track — Submitted to NeurIPS 2025 Datasets and Benchmarks Track_

### Official Review · Reviewer_G65A · 2025-07-01

**Rating:** 4
**Confidence:** 3

**Summary:**

The paper presents CFDLLMBench, a benchmark suite designed to evaluate Large Language Models (LLMs) in automating numerical experiments in Computational Fluid Dynamics (CFD). The benchmark consists of three components—CFDQuery, CFDCodeBench, and FoamBench—which assess LLMs on graduate-level CFD knowledge, numerical and physical reasoning, and workflow implementation. The authors provide an easy-to-use framework for evaluating LLM-generated code based on execution success, solution accuracy, and numerical convergence, offering a valuable foundation for advancing LLM applications in scientific computing.

**Additional Feedback:**

* In Figure 1 (FoamBench), the figure above the caption ("Create case files (...)") appears upside-down. Please correct this.

* In Figure 8 of the appendix, the content is too small to read comfortably. Please consider enlarging it for clarity.

* In the appendix, Figure 9's horizontal axes are not ordered consistently. A unified order would enhance readability.

**Dataset Code Accessibility:**

Yes

**Dataset Code Comments:**

* While the GitHub READMEs are well-written, their formatting is inconsistent—likely due to being authored by different contributors. Standardizing the README format would improve usability.

* A download script for the dataset is recommended to streamline data acquisition; manually downloading from Kaggle can be time-consuming.

* The license status of the dataset sources is unclear. Clarification on licensing or copyright issues would be beneficial.

**Ethical Considerations:**

Yes, there are ethics concerns that require attention by the authors

**Final Justification:**

Thank you to the authors for their response to my comments. I find the responses satisfactory and will maintain my current positive evaluation.

**Limitations Weaknesses:**

Major Issues:

* W1: As noted in the "Limitations" section, the absence of human performance baselines is a significant flaw. For example, expert evaluations on FoamBench would help validate the used prompt quality; otherwise, it's unclear whether LLM failure stems from prompt deficiencies or LLM model limitations.

* W2: The expert contributors appear biased toward early-career researchers (undergraduates, PhD students, postdocs). It would strengthen the dataset quality assessment to include more senior experts (ideally with over 10 years of CFD experience). Currently, responsibility for dataset validation rests mainly with a single professor, which is insufficient and unrealistic to assume a perfect management of the quality of all the datasets and prompts.

Minor Issues:

* W3: The choice of NMSE thresholds (<10%, 30%) for defining M_NMSE needs justification. Are these values reflective of expected variations due to different but valid numerical schemes? Please provide a quantitative basis showing that, for example, a 10% difference is within the natural range of such discrepancies.

* W4: For CFDQuery, allowing LLMs to access external sources (e.g., web or scientific literature) would better reflect real-world usage, especially in settings like DeepResearch.

**Strengths Contributions:**

* The codebase is well-prepared and thoroughly documented, with plans to make it publicly available on GitHub.

* The dataset is accessible via Kaggle, although the download process is somewhat cumbersome (see below for suggestions).

* The benchmark evaluates a variety of LLMs, including more advanced agent-based approaches, offering a comprehensive comparison of capabilities.

---

> ### Author Rebuttal · Authors · 2025-07-28
>
> We thank the reviewer for their positive feedback and for recognizing the value of CFDLLMBench in advancing LLM applications in scientific computing. We are pleased that the reviewer appreciated the well-prepared codebase and the comprehensive evaluation of various LLMs, including agent-based approaches. We address the concerns as follows.
>
> >**W1. As noted in the "Limitations" section, the absence of human performance baselines is a significant flaw. For example, expert evaluations on FoamBench would help validate the used prompt quality; otherwise, it's unclear whether LLM failure stems from prompt deficiencies or LLM model limitations.**
>
> A1. As noted in our limitations section, we agree that including a human baseline would strengthen the benchmark. However, even without explicit human results, approximate human performance can be reasonably inferred by domain experts. For example, **closed-source LLMs achieve 85–90% on CFDQuery, comparable to or above typical human engineers in a closed-book setting. In contrast, on CFDCodeBench, humans can reliably achieve near-perfect performance, while the best LLM reaches only 14%, showing a clear gap. For FoamBench, even the best agent reaches 34% success, far below what a trained OpenFOAM user would easily achieve**.
>
> Regarding prompt quality, we performed a quick experiment on FoamBench Advanced. The original prompts were refined through multiple rounds of disambiguation using the O3 reasoning model, with human oversight. Five prompt variants were tested with Claude Sonnet 3.5 in a zero-shot setting. For the best performing variant the **success ratio increased only marginally (from 0.007 to 0.012), indicating that the original prompts were already well-structured**. A more comprehensive study will be included in the camera-ready version.
>
> | Dataset            | M\_exec | M\_struct | M\_file | M\_NMSE |   Success Rate |
> | ------------------ | ------- | --------- | ------- | ------------ | ------------ |
> | FoamBench Advanced | 0.034   | 0.769     | 0.588   | 0.012 | 0.012 |
>
> *By contrast, components such as RAG and the Reviewer have a much larger impact, boosting Claude Sonnet 3.5’s success to ~25% (see Table 2 and Table 4). These findings suggest that the observed LLM limitations stem from reasoning and tool-use challenges, not prompt deficiencies. We agree that incorporating human baselines and more extensive prompt optimization are valuable future directions and plan to include them by camera ready deadline.*
>
> >**W2. The expert contributors appear biased toward early-career researchers (undergraduates, PhD students, postdocs). It would strengthen the dataset quality assessment to include more senior experts (ideally with over 10 years of CFD experience). Currently, responsibility for dataset validation rests mainly with a single professor, which is insufficient and unrealistic to assume a perfect management of the quality of all the datasets and prompts.**
>
> A2. **CFDLLMBench is intended as a live, evolving resource, and we agree that ongoing involvement from the broader CFD community, including senior experts, will further enhance its quality**. For the current release, dataset creation primarily involved early-career researchers (PhD students and postdocs), who are actively engaged in CFD research. **We intentionally focused on fundamental, generic CFD problems and workflows well within their expertise, rather than advanced frontier topics requiring extensive senior judgment**. More challenging, research-level problems will be included in future expansions. As CFDLLMBench continues to evolve, we plan to actively invite contributions and validations from senior CFD professionals to ensure long-term robustness and continuous improvement.
>
> >**W3. The choice of NMSE thresholds (<10%, 30%) for defining M_NMSE needs justification. Are these values reflective of expected variations due to different but valid numerical schemes? Please provide a quantitative basis showing that, for example, a 10% difference is within the natural range of such discrepancies.**
>
> A3. These values were not chosen arbitrarily but are grounded in **engineering practice** and further supported by an **empirical sensitivity analysis**.
>
> 1. **Engineering Practice**: **CFD engineering practice commonly follows the thumb rule that an NMSE below 10% indicates an accurate simulation, while errors above 30% mark the upper limit for accuracy**.
> In CFD and related engineering fields, an NMSE (or relative error) below approximately 10%, typically resulting from well-configured numerical setups, is widely regarded as indicative of an accurate and reliable simulation. Conversely, errors exceeding 30% are generally considered practically unacceptable when validating simulations against numerical ground truth. These brackets are routinely used in both academic validation studies and industrial verification.
>
> 2. **Empirical Sensitivity Analysis**: **Validate threshold choices by altering the upper and lower limit and measuring their impact on success rates**.
>  To further justify our choice, we conducted a sensitivity study by varying the thresholds and observing their effect on both mean NMSE score and the true success rate.
>
> Table A. Mean NMSE scores with varying lower bounds (upper bound fixed at 30%).
> | Lower Bound | Mean NMSE Score |
> | ----------- | --------------- |
> | 1%          | 0.3909          |
> | 5%          | 0.4000          |
> | **10%**         | **0.4273**          |
> | 15%         | 0.4318          |
>
> **The table shows a clear progression, with the strongest gain observed at 10%, beyond which increase is marginal.**
>
> Table B. Sensitivity of true success rate to different lower NMSE cutoffs (upper bound fixed at 30%).
> | Lower NMSE Bound | True Success Rate |
> | ---------------- | ----------------- |
> | 1%               | 26.4%             |
> | 5%               | 28.2%             |
> | **10%**              | **33.6%**             |
> | 15%              | 34.5%             |
>
> Table C. Mean NMSE scores with varying upper bounds (lower bound fixed at 10%).
> | Upper Bound | Mean NMSE Score |
> | ----------- | --------------- |
> | 0.25        | 0.4045          |
> | **0.30**        | **0.4273**          |
> | 0.40        | 0.4955          |
> | 0.45        | 0.5045          |
>
> **It can be seen that beyond 30%, the metrics becomes overly accommodative and can include edge cases.**
>
> *The combination of domain-standard brackets (10% and 30%) and our sensitivity analysis demonstrates that 10% is the optimal cutoff for accurately identifying correct simulations, while 30% serves as a natural upper limit for defining unacceptable solutions. These thresholds align with established CFD practices and ensure that the metric remains interpretable and meaningful.*
>
> >**W4. For CFDQuery, allowing LLMs to access external sources (e.g., web or scientific literature) would better reflect real-world usage, especially in settings like DeepResearch.**
>
> A4. **Allowing LLMs to access external sources, such as the web or scientific literature, could indeed better reflect real-world usage scenarios. However, introducing retrieval during evaluation could confound the conclusions, as it becomes challenging to disentangle the model’s reasoning capability from its ability to search and integrate external information**. For this reason, we focused the current benchmark on evaluating the **models’ intrinsic knowledge and reasoning without external assistance**. We acknowledge that incorporating retrieval is potentially beneficial for understanding LLM performance in more realistic scenarios but it is orthogonal to our current study. Moreover, even if LLM agents with retrieval capabilities perform better, this does not invalidate our findings; rather, **it highlights the complementary role of agentic frameworks** in enhancing model performance, a direction we view as promising for future work.
>
> >**Dataset and Code Comments**
>
> We have updated the README to be more consistent and included the dataset download and save script. The license status of the dataset is set to be BSD‑3.
>
> >**Additional Feedback**
>
> We thank the reviewer for carefully pointing out these presentation issues. We apologize for the oversight. These will be corrected and refined in the camera-ready version.

---

### Official Review · Reviewer_zmgJ · 2025-07-03

**Rating:** 3
**Confidence:** 2

**Summary:**

This paper introduces CFDLLMBench, a benchmark for assessing LLMs in Computational Fluid Dynamics (CFD) across three key competencies: (1) graduate-level CFD knowledge via multiple-choice questions (CFDQuery), (2) code generation and reasoning in solving PDEs found in CFD(CFDCodeBench), and (3) real-world OpenFOAM simulation workflow implementation (FoamBench).  Each component is paired with defined and task-related metrics. Results indicate clear gaps between current LLM abilities in general science Q&A and the more demanding real-world CFD simulation tasks

**Dataset Code Accessibility:**

Yes

**Dataset Code Comments:**

The dataset is accessible and has passed the test.

**Ethical Considerations:**

No, there are no or only very minor ethics concerns

**Final Justification:**

Resolved or Partially Resolved Issues
- Q1:  The rebuttal does not provide new human evaluation results but does offer quantitative estimates of expected human performance per task based on domain expertise.
- Q2 Authors justify the zero-shot design as an intentional choice to measure intrinsic model capability without bias from hints, noting that few-shot prompting gave only marginal gains and could artificially inflate scores. This rationale addresses the original concern’s reasoning, though empirical evidence for the “marginal gains” claim is brief.
- Q3:  Authors agree with the concern and argue that reasoning steps are transferable, while acknowledging practical barriers . They clarify that their benchmark methodology is adaptable. This addresses the conceptual part of the concern, though no cross-platform experiments are shown.
- Q4: Authors explain a multi-stage filtering process to remove trivially solvable questions and commit to adopting transformation techniques to reduce memorization risk in future iterations. This partially mitigates the concern, but no direct contamination checks were done.

Remaining Unresolved Issues
- No actual measured human results are included in this version.
- While the rationale for zero-shot is plausible, there is limited quantitative evidence on alternative prompting impacts beyond a qualitative statement.
- no validation outside OpenFOAM is performed.
- Mitigation steps are heuristic; lack of empirical overlap analysis leaves uncertainty about true contamination risk.

The rebuttal provides reasonable clarifications and commits to future improvements, which increase confidence in the design choices and dataset quality. However:
- Key limitations remain unaddressed in the current version (measured human baseline, empirical contamination check, demonstration of generalization beyond OpenFOAM).
- The additional explanations strengthen the paper’s transparency but do not substantially change the technical completeness or scope of contributions in this submission cycle.

Given these points, my original score still reflects the paper’s merits and current limitations.

**Limitations Weaknesses:**

- The paper acknowledges the absence of a human baseline. This makes it hard to calibrate LLM performance, especially given that some tasks may be highly specialized or easier/harder than assumed. A qualitative or limited quantitative baseline would provide a powerful anchor to interpret whether a 34% success rate is poor or promising.

- The use of a standard zero-shot prompt for CFDQuery and CFDCodeBench may not fully reflect the models' potential.

- While agentic frameworks improve performance, it’s unclear how well these agent setups generalize beyond FoamBench or to other simulation software. My educated guess is that the agentic frameworks should perform better when well configured.

- The creation of CFDQuery is partly from online material. It is possible that they are exposed to LLMs already. There is no explicit discussion of the steps taken to ensure that LLMs have not seen test questions or tasks during pretraining.

**Strengths Contributions:**

- The holistic approach in the paper addresses a critical gap in existing scientific LLM benchmarks.

- All tasks are constructed or authenticated by experts in CFD, with issues of domain authenticity and coverage clearly addressed.

- The inclusion of results on LLMs acting within agentic frameworks (MetaOpenFOAM, Foam-Agent), and the corresponding quantitative impact is insightful and relevant for workflow automation research.

---

> ### Author Rebuttal · Authors · 2025-07-28
>
> We thank the reviewer for their positive assessment of our work. We are encouraged that they recognized CFDLLMBench as addressing a critical gap in scientific LLM benchmarks through its holistic design, expert-authenticated tasks, and comprehensive coverage of CFD competencies. We also appreciate the acknowledgment of our inclusion of agentic frameworks and their quantitative impact, which we believe provides valuable insights for advancing workflow automation research in CFD. We address the concerns as follows.
>
> >**Q1. The paper acknowledges the absence of a human baseline. This makes it hard to calibrate LLM performance, especially given that some tasks may be highly specialized or easier/harder than assumed. A qualitative or limited quantitative baseline would provide a powerful anchor to interpret whether a 34% success rate is poor or promising.**
>
> A1. As noted in our limitations section, we agree that including a human baseline would strengthen the benchmark. However, even without explicit human results, our benchmark still provides a meaningful and rigorous evaluation of current LLM capabilities as the tasks are designed such that **approximate human performance can be reasonably inferred by domain experts and compared against the capabilities of human CFD practitioners**.
>
> 1. **CFDQuery (Conceptual Knowledge)**: The best LLMs (closed-source) score 85–90% on graduate-level CFD questions. **This is on par with, or even above, what a typical CFD engineer would achieve under a closed-book setting**, as humans rely on references for such a wide knowledge scope.
>
> 2. **CFDCodeBench (Numerical Reasoning & Code Generation)**: The top LLM scores only 14% on simple PDE solver tasks (e.g., diffusion, Burgers). **A CFD trained graduate student can reliably solve these with high accuracy by writing a small script or reusing existing templates, highlighting a large gap between LLM memorization and genuine reasoning/coding.**
>
> 3. **FoamBench (Workflow Automation)**: Even with an agentic setup, the best model achieves only 34% success on standard OpenFOAM tutorial cases. **A CFD engineer familiar with OpenFOAM would easily solve most of these tasks, showing that current LLMs struggle with decomposition and physics-driven workflow generation**.
>
> *Conducting a comprehensive human evaluation across all components would require substantial organization, time, and expert involvement. We have initiated additional studies to quantify human performance and aim to include the results in the camera-ready version*.
>
> >**Q2. The use of a standard zero-shot prompt for CFDQuery and CFDCodeBench may not fully reflect the models' potential.**
>
> A2.  **Our choice to use a single-pass zero-shot setting for CFDQuery and CFDCodeBench was intentional, as these components aim to evaluate the model’s intrinsic knowledge and reasoning ability without relying on external feedback, iterative refinement, or hints embedded in the prompt**. We agree that few-shot or advanced prompting techniques can improve model outputs in many domains. **In our experiments, few-shot prompts and structured hints provided only marginal improvements while introducing bias, as models often reused supplied code fragments without demonstrating genuine problem-solving ability**. We designed the benchmark to follow a curriculum from simple to complex. CFDCodeBench focuses on basic PDEs in 1D and 2D, establishing a foundational baseline for model capability, tasks solvable by a graduate student but still challenging for current LLMs. In contrast, FoamBench explicitly evaluates agentic workflows where iterative refinement with a reviewer and RAG is integral to the task.
>
> We acknowledge that incorporating self-refinement strategies for CFDQuery and CFDCodeBench is an interesting future direction. Such extensions, along with exploring optimized prompts, will complement the current zero-shot.
>
> >**Q3. While agentic frameworks improve performance, it’s unclear how well these agent setups generalize beyond FoamBench or to other simulation software. My educated guess is that the agentic frameworks should perform better when well configured.**
>
> A3. **Yes, we agree that evaluating how well agentic frameworks generalize beyond FoamBench and to other CFD software is an important consideration**. Our study focuses on OpenFOAM because it is one of the most widely used open-source CFD tools in academia and industry. FoamBench itself is comprehensive, containing both basic and advanced cases that cover a diverse range of CFD workflows, probing agentic reasoning across multiple scenarios.
>
> We agree with the reviewer’s observation that well-configured agentic frameworks should generalize to other simulation platforms, as the underlying reasoning steps are similar. However, extending to widely used commercial softwares faces practical barriers, including GUI-driven workflows, license restrictions, and proprietary processes.
>
> Finally, we emphasize that FoamBench benchmarks **agentic workflows specifically designed for OpenFOAM**. While our work focuses on OpenFOAM, the benchmarking cases and methodology is general and can be adapted to evaluate other agentic frameworks built around different CFD software or tools.
>
>
> >**Q4. The creation of CFDQuery is partly from online material. It is possible that they are exposed to LLMs already. There is no explicit discussion of the steps taken to ensure that LLMs have not seen test questions or tasks during pretraining.**
>
> A4. **Yes, some of the questions in CFDQuery originate from online material, which raises the possibility that LLMs may have seen parts of the dataset during pretraining. While it is practically impossible to fully eliminate such contamination, we took steps to minimize its impact**. Specifically, we began with a pool of 300 curated questions for CFDQuery and conducted multiple iterations with several LLMs, removing questions that were trivially solved by all models. After this iterative filtering, the remaining questions underwent expert review to ensure correctness, clarity, and coverage (similar to Humanity’s last Exam). Through this process, the dataset was further refined to the final set of 90 questions.
>
> Inspired by LLM-SRBench (Shojaee et al., 2025), which mitigates contamination by transforming known equations and introducing synthetic terms, we plan to adopt a similar strategy in future iterations of our benchmark. This will include paraphrasing, problem transformation, and introducing novel variations to make the tasks less susceptible to memorization.

---

> > ### Comment · Reviewer_zmgJ · 2025-08-06
> >
> > Thank you for your thoughtful rebuttal. I appreciate your clarifications and have decided to maintain my original score.

---

> > > ### Author Response · Authors · 2025-08-06
> > >
> > > Thank you very much for your detailed review and for engaging with our rebuttal. We appreciate your thoughtful feedback and are grateful for the time you've spent assessing our submission. If there are any remaining concerns or areas where additional clarification would be helpful, we would be happy to provide further information or elaboration before the final decision.
> > >
> > > Please let us know if there's anything specific you would like us to address further.

---

### Official Review · Reviewer_1Uxz · 2025-07-03

**Rating:** 4
**Confidence:** 4

**Summary:**

This paper presents a dataset for testing and validating the usage of LLMs on CFD related tasks. The dataset contains three components: CFDQuery (knowledge of numerical analysis), CFDCodeBench (ability to understand a CFD problem description, then programming it, and present results) and FoamBench (solving more complex problems using OpenFOAM). FoamBench also makes use of agentic models to execute various tasks. The dataset has been put together by six domain experts.

CFDQuery is a set of 108 questions along with multiple choices for answers where one of them is the correct answer. CFDCodeBench is a set of 24 prompts describing a CFD problem that asks the LLM to write a Python program to solve the problem. And FoamBench is a set of 126 prompts (and corresponding supporting files) for generating the correct configuration files that can then be input to OpenFOAM for solving complex CFD problems. 110 of these are "basic" and the rest are "advanced." Here the tasks could be such as creating the geometry, or creating the mesh, or choosing the correct turbulence model etc.

The authors choose a total of six LLMs (five closed-source, one open-source), and test and benchmark them against these three dataset / tasks. CFDQuery is tested with a "success / failure" outcomes. CFDCodeBench is tested by three components: (i) code executability, (ii) relative numerical error, (iii) numerical convergence. Evaluation on FoamBench is done through (i) executability, (ii) file / folder organization, (iii) file similarity, (iv) relative error.

**Dataset Code Accessibility:**

Yes

**Ethical Considerations:**

No, there are no or only very minor ethics concerns

**Final Justification:**

In one way, the contributions in this dataset are commendable, because the nature of the dataset is such that it requires high level expertise in CFD methods and programming. On the other hand, the dataset is not vast enough to be a benchmarking tool. However, it could be good resource in research and learning.

My impression of this paper remains the same, as in, I am still on the borderline but leaning toward accept, and so I will maintain this score.

**Limitations Weaknesses:**

* The dataset is commendable but they are still limited. For example, there is no 3D case (hoping that I have not missed something), thereby limiting the most important applications. This comment may or may not apply to CFDCodeBench since the python programs on simple 3D geometries often need only minor adjustments over the 2D ones. However, it does matter in case of FoamBench since the meshing step in 3D is a non-trivial, and highly time-consuming and iterative process.
* The conclusion in the paper is that most of the closed source LLMs are performing on roughly at the same level, and better than the one open-source LLM tested here. Beyond this information, I do not see much utility of this dataset for CFD practitioners. The paper does not provide any insight on how to use theses models with improved outcomes.

**Strengths Contributions:**

* Valuable dataset for testing / benchmarking / comparing LLMs on CFD related tasks. Such a dataset takes a significant amount of time and effort, not to mention the expertise. Thus, this is a valuable contribution.
* Both CFDCodeBench and FoamBench feature a varied set of problems that can be encountered in CFD analysis (1D and 2D). Many of these problems are standard.
* The evaluation criteria that are used are simple and intuitive.

---

> ### Author Rebuttal · Authors · 2025-07-28
>
> We thank the reviewer for their thoughtful feedback and for recognizing our contributions, particularly highlighting the substantial time, effort, and domain expertise required to assemble this dataset. We appreciate the reviewer acknowledging the diversity and practical relevance of problems in CFDCodeBench and FoamBench, as well as noting that our evaluation criteria are intuitive and effective for benchmarking LLM performance on Computational Fluid Dynamics tasks. We address the concerns as follows.
>
> >**Q1. The dataset is commendable but they are still limited. For example, there is no 3D case (hoping that I have not missed something), thereby limiting the most important applications. This comment may or may not apply to CFDCodeBench since the python programs on simple 3D geometries often need only minor adjustments over the 2D ones. However, it does matter in case of FoamBench since the meshing step in 3D is a non-trivial, and highly time-consuming and iterative process.**
>
> A1. We agree that inclusion of 3D cases in FoamBench is important.
> **FoamBench does include several 3D tutorial cases:squareBend, shallowWaterWithSquareBump, and damBreakWithObstacle**. These tasks require the agentic system to construct valid 3D geometries, generate meshes, and execute simulations using OpenFOAM. Our benchmark evaluates whether  LLMs can generalize from tutorial cases to correctly set up new complex 3D meshes using OpenFOAM’s tools, including **blockMesh and snappyHexMesh**. We will emphasize this point more explicitly in the camera-ready version of the manuscript.
>
> **We also agree with the reviewer that CFDCodeBench does not necessitate 3D meshing, as its tasks involve simple domains where extending to 3D requires only minor code adjustments, and no specialized meshing utilities are used**. Thus, while CFDCodeBench remains intentionally 1D/2D, FoamBench already benchmarks the more advanced aspects of 3D geometry creation, meshing, and workflow automation under an agentic framework with RAG.
>
> >**Q2. The conclusion in the paper is that most of the closed source LLMs are performing on roughly at the same level, and better than the one open-source LLM tested here. Beyond this information, I do not see much utility of this dataset for CFD practitioners. The paper does not provide any insight on how to use theses models with improved outcomes.**
>
> A2. **We take this opportunity to clarify the broader utility of CFDLLMBench. While our primary goal is to benchmark LLMs rather than improve them, this dataset offers several concrete benefits to both the AI for Science community and CFD practitioners**:
> 1. **Evaluation of LLMs and Agentic Frameworks**: The benchmark provides a rigorous way to assess not only the raw reasoning ability of LLMs but also their performance when embedded in **agentic frameworks**. Researchers can use it to identify weaknesses in planning, reasoning, or tool usage.
>
> 2. **Playground for Finetuning and Model Development**: CFD practitioners and AI researchers can use the dataset to **finetune smaller open-source models** or develop task-specific LLMs that perform better on CFD tasks. The diverse and challenging nature of the dataset ensures that models trained on it generalize beyond toy problems.
>
> 3. **Code Generation as a Learning Tool**: For **CFD education**, CFDCodeBench tasks mirror how humans learn by writing code, tuning parameters, and analyzing results. If an LLM can generate correct engineering code, it can serve as a valuable teaching assistant, helping students quickly test ideas and understand PDEs. Conversely, poor code generation highlights areas where models need improvement.
>
> 4. **FoamBench for Productivity Gains**: OpenFOAM practitioners often start from tutorials and iteratively adapt them. FoamBench mimics this workflow, making it a **practical benchmark for evaluating and improving productivity**. Success on FoamBench suggests an LLM can accelerate case setup and troubleshooting, directly benefiting engineers.
>
> 5. **Foundation for Future Research**: By setting a challenging baseline, CFDLLMBench serves as a **playground for future model development**. Models that perform well on this benchmark are likely to have lower hallucination rates and better domain grounding, making them more reliable in real-world engineering workflows.
>
> By exposing gaps (e.g., weak reasoning in code generation, poor configuration of OpenFOAM workflows), this benchmark lays the groundwork to measure future improvements in both LLM design and their integration into CFD pipelines. In a separate work, this benchmark provided insights to improve the design of agentic frameworks.

---

### Official Review · Reviewer_WrUQ · 2025-07-06

**Rating:** 5
**Confidence:** 4

**Summary:**

This paper introduces a novel benchmark to evaluate LLMs abilities on Computational Fluid Dynamics (CFD) tasks. The benchmark consists of three tasks/datasets (CFDQuery, CFDCodeBench, OpenBench), one is evaluating graduate level knowledge through objective questions, other is evaluating numerical abilities for CFD and the third being generating/implementing workflows for solving specific CFD problems.

Building on previous works and extending to more real-world scenarios, this establishes a strong benchmark dataset for evaluating CFD capabilities of LLMs.

**Additional Feedback:**

*Questions*

In section 3.2 Quality Assessment: Was the review and revision done by a different set of experts than those mentioned in the _Experts Contribution_ section

Why are all 4 metrics not used in the success rate calculation for the FoamBench evaluation? If not file similarity, file structure seems a metric which directly affects the correctness of the task, which should have been incorporated in the metric.

Appendix A.3.1 does mention of RAG, how is the RAG being currently implemented, what is the corpus and what is the retriever?

**Dataset Code Accessibility:**

Yes

**Dataset Code Comments:**

The entire code for the dataset evaluation is provided (for the reviewers currently, I have glanced through the dataset and supplementary material) and will be publicly released with appropriate license. The section 3.4 of the paper discusses this in detail.

**Ethical Considerations:**

No, there are no or only very minor ethics concerns

**Final Justification:**

I have read through all the reviews, rebuttals and responses. For my questions on prompt-engineering and RAG, the authors ran some experiments during the rebuttal phase to validate their claims. For my query on choice of metrics, detailed analysis has been provided to substantiate the buckets for NMSE errors. The authors have also initiated a study to quantify human performance and aim to include it. Considering all these I would increase my score by a point.

**Limitations Weaknesses:**

The authors have included a limitations section and have highlighted that the benchmark doesn’t have any human baselines as it is very difficult to fairly evaluate human performance on these tasks due to variance in knowledge, skill, speed etc.

The metrics defined seem a bit arbitrary which makes it difficult to interpret the scores. If we look at equation *(1)* In section 3.3, how do the authors arrive at buckets of 10% and 30% is not very clear.

A limitation of this work is the lack of prompt engineering as the authors have mentioned. I believe unless a detailed analysis with the most optimized prompt is not done, it would be difficult to gauge the true reflection of the model's abilities.

**Strengths Contributions:**

The authors provide an elaborate discussion on the related work and places itself well among the existing literature landscape.

A holistic dataset which evaluates factual understanding/retrieval, numerical/mathematical abilities and complex reasoning/planning for CFD tasks. Most previous works evaluate specific abilities or on relatively smaller/simpler problems.

Extremely detailed descriptions on experimental configurations, setup, model hyperparameters and ablations have been provided in the appendices.

---

> ### Author Rebuttal · Authors · 2025-07-28
>
> We thank the reviewer for their thoughtful and constructive feedback. We are encouraged that they highlighted several strengths of our work, including its novel benchmark design, comprehensive evaluation of multiple CFD abilities (knowledge, numerical reasoning, and workflow automation), and clear positioning within the existing literature. We also appreciate that they recognized the holistic nature of our benchmark, the detailed discussion of related work, and the extensive experimental configurations and ablations provided. We address the concerns as follows.
>
>
>
> >**Q1. The authors have included a limitations section and have highlighted that the benchmark doesn’t have any human baselines as it is very difficult to fairly evaluate human performance on these tasks due to variance in knowledge, skill, speed etc.**
>
> A1. As noted in our limitations section, we agree that including a human baseline would strengthen the benchmark. However, even without explicit human results, our benchmark still provides a meaningful and rigorous evaluation of current LLM capabilities as the tasks are designed such that **approximate human performance can be reasonably inferred by domain experts and compared against the capabilities of human CFD practitioners**.
>
> 1. **CFDQuery (Conceptual Knowledge)**: The best LLMs (closed-source) score 85–90% on graduate-level CFD questions. **This is on par with, or even above, what a typical CFD engineer would achieve under a closed-book setting**, as humans rely on references for such a wide knowledge scope.
>
> 2. **CFDCodeBench (Numerical Reasoning & Code Generation)**: The top LLM scores only 14% on simple PDE solver tasks (e.g., diffusion, Burgers). **A CFD trained graduate student can reliably solve these with high accuracy by writing a small script or reusing existing templates, highlighting the gap between LLM memorization and genuine reasoning/coding.**
>
> 3. **FoamBench (Workflow Automation)**: Even with an agentic setup, the best model achieves only 34% success on standard OpenFOAM tutorial cases. **A CFD engineer familiar with OpenFOAM would easily solve most of these tasks, showing that current LLMs struggle with decomposition and physics-driven workflow generation**.
>
> *Conducting a comprehensive human evaluation across all components would require substantial organization, time, and expert involvement. We have initiated additional studies to quantify human performance and aim to include the results in the camera-ready version*.
>
>
>
> >**Q2. The metrics defined seem a bit arbitrary which makes it difficult to interpret the scores. If we look at equation (1) In section 3.3, how do the authors arrive at buckets of 10% and 30% is not very clear.**
>
> A2. These values were not chosen arbitrarily but are grounded in **engineering practice** and further supported by an **empirical sensitivity analysis**.
>
> 1. **Engineering Practice**: **CFD engineering practice commonly follows the thumb rule that an NMSE below 10% indicates an accurate simulation, while errors above 30% mark the upper limit for accuracy**.
> In CFD and related engineering fields, an NMSE (or relative error) below approximately 10%, typically resulting from well-configured numerical setups, is widely regarded as indicative of an accurate and reliable simulation. Conversely, errors exceeding 30% are generally considered practically unacceptable when validating simulations against numerical ground truth. These brackets are routinely used in both academic validation studies and industrial verification.
>
> 2. **Empirical Sensitivity Analysis**: **Validate threshold choices by altering the upper and lower limit and measuring their impact on success rates**.
>  To further justify our choice, we conducted a sensitivity study by varying the thresholds and observing their effect on both mean NMSE score and the true success rate.
>
> Table A. Mean NMSE scores with varying lower bounds (upper bound fixed at 30%).
> | Lower Bound | Mean NMSE Score |
> | ----------- | --------------- |
> | 1%          | 0.3909          |
> | 5%          | 0.4000          |
> | **10%**         | **0.4273**          |
> | 15%         | 0.4318          |
>
> Table B. Sensitivity of true success rate to different lower NMSE cutoffs (upper bound fixed at 30%).
> | Lower NMSE Bound | True Success Rate |
> | ---------------- | ----------------- |
> | 1%               | 26.4%             |
> | 5%               | 28.2%             |
> | **10%**              | **33.6%**             |
> | 15%              | 34.5%             |
>
> **The table shows a clear progression, with the strongest gain observed at 10%, beyond which increase is marginal.**
>
> Table C. Mean NMSE scores with varying upper bounds (lower bound fixed at 10%).
> | Upper Bound | Mean NMSE Score |
> | ----------- | --------------- |
> | 0.25        | 0.4045          |
> | **0.30**        | **0.4273**          |
> | 0.40        | 0.4955          |
> | 0.45        | 0.5045          |
>
> **It can be seen that beyond 30%, the metrics becomes overly accommodative and can include edge cases.**
>
> *The combination of domain-standard brackets (10% and 30%) and our sensitivity analysis demonstrates that 10% is the optimal cutoff for accurately identifying correct simulations, while 30% serves as a natural upper limit for defining unacceptable solutions. These thresholds align with established CFD practices and ensure that the metric remains interpretable and meaningful.*
>
>
>
> >**Q3. A limitation of this work is the lack of prompt engineering as the authors have mentioned. I believe unless a detailed analysis with the most optimized prompt is not done, it would be difficult to gauge the true reflection of the model's abilities.**
>
> A3. We have not conducted an extensive ablation on prompt engineering. While prompt optimization is typically useful, **our experience showed that advanced methods like RAG and Reviewer tools have a much larger impact**. Nonetheless, all models were evaluated under identical prompt settings, ensuring fairness. We acknowledge that further validation on prompt engineering would strengthen the analysis.
>
> To partially address this during the rebuttal phase, we conducted a prompt engineering experiment on FoamBench Advanced. Human-authored prompts were iteratively refined using the O3 reasoning model, with five validated variants tested on Claude Sonnet 3.5 in a zero-shot setting (without RAG or Reviewer). **The best variant improved success rate only marginally (0.007 to 0.012), indicating the original prompts were already effective**. A more comprehensive study will be included in the camera-ready version.
>
> Table D. FoamBench Advanced Metrics.
> | Dataset            | M\_exec | M\_struct | M\_file | M\_NMSE |   Success Rate |
> | ------------------ | ------- | --------- | ------- | ------------ | ------------ |
> | FoamBench Advanced | 0.034   | 0.769     | 0.588   | 0.012 | 0.012 |
>
> By contrast, components such as **RAG and the Reviewer have shown a far greater impact, boosting Claude Sonnet 3.5’s success to ~0.25**. These ablations are detailed in Table 2 (main text) and Table 4 (appendix).
>
>
>
> >**Q4. In section 3.2 Quality Assessment: Was the review and revision done by a different set of experts than those mentioned in the Experts Contribution section**
>
> A4. **Yes, the quality assessment was conducted by experts who were not directly involved in the initial dataset generation**. All datasets were curated by a team of graduate students and subsequently reviewed by an independent panel consisting of a postdoctoral researcher and a faculty member, who ensured correctness, clarity, and coverage.
>
>
>
> >**Q5. Why are all 4 metrics not used in the success rate calculation for the FoamBench evaluation? If not file similarity, file structure seems a metric which directly affects the correctness of the task, which should have been incorporated in the metric.**
>
> A5. **We highlight in the main paper that file and structural similarity metrics are not reliable indicators of success in practical CFD scenarios. In practice, outcome correctness, whether the generated simulation accurately reproduces the numerical ground truth, is far more critical than the resemblance of input files/folders to reference templates**. As noted in Table 2, the best-performing model (Claude Sonnet 3.5) achieves high file similarity ($M_{\text{file}} \approx 0.8$) and structural similarity ($M_{\text{struct}} \approx 0.9$), yet its true success rate is only 0.33, clearly demonstrating this disconnect. Therefore, as discussed in Section 3.3, we use execution and numerical accuracy (via NMSE) as the decisive criteria for success. File- and structure-based similarity metrics remain as classic/standard diagnostic indicators but are not reliable enough to be part of the success rate calculation.
>
>
>
> >**Q6. Appendix A.3.1 does mention of RAG, how is the RAG being currently implemented, what is the corpus and what is the retriever?**
>
> A6. **We follow the existing RAG implementations of Foam-Agent[2] and MetaOpenFoam[1]**.
> The corpus consists of a database built from OpenFOAM tutorials, containing case names, categories, directory structures, and complete file contents. Each case is encoded using ChatGPT's embedding model. The retriever uses FAISS to find the most similar case based on embedding similarity between the user query and stored cases. The retrieved case is then incorporated into the LLM context as additional information. For additional information please refer to the manuscripts of these frameworks.
>
> 1. Chen, Y., Zhu, X., Zhou, H., and Ren, Z. Metaopenfoam: an llm-based multi-agent framework for cfd. arXiv preprint arXiv:2407.21320, 2024.
> 2.  Yue, L., Somasekharan, N., Cao, Y., and Pan, S. Foam-agent: Towards automated intelligent cfd workflows. arXiv preprint arXiv:2505.04997, 2025.

---

### Author Response · Authors · 2025-08-05
**Overall Author Comment**

We thank all reviewers for their thoughtful and constructive evaluations. Across the reviews, there was consensus that **CFDLLMBench makes a timely and significant contribution by introducing the first of its kind holistic benchmark to evaluate LLMs on Computational Fluid Dynamics (CFD) tasks**. The benchmark spans three complementary components:
- **CFDQuery** – tests graduate-level CFD knowledge through multiple-choice questions.
- **CFDCodeBench** – assesses numerical reasoning and code generation for PDE solvers.
- **FoamBench** – evaluates real-world CFD workflow automation using OpenFOAM and agentic LLM frameworks.

Overall, the reviews acknowledge that **CFDLLMBench is a technically solid and impactful benchmark that addresses a pressing need for evaluating LLMs in scientific domains**. The benchmark lays a foundation for future research in **LLM reasoning, agentic workflows, and AI for Science**, and has strong potential to influence both the AI and CFD communities.

**Reviewers recognized several strengths**:
- The benchmark fills a critical gap in scientific AI evaluation by covering factual knowledge, numerical reasoning, and workflow planning [WrUQ, zmgj].
- It is a valuable and labor-intensive contribution that required significant domain expertise to build. Beyond its scale, the benchmark stands out for its diversity by covering a wide spectrum of realistic CFD scenarios, from standard 1D/2D problems to complex workflows, making it both practically relevant and broadly applicable for evaluating LLM capabilities [1Uxz].
- All tasks were constructed and validated by domain experts, with detailed configurations, ablations, and clear positioning in the literature [zmgj].
- The inclusion of agentic frameworks (e.g., RAG, Reviewer) provided valuable insights into workflow automation research [zmgj, G65A].
- The dataset, code, and evaluation pipeline are well-documented, reproducible, and will be released with a permissive license [G65A].

**Common Concerns Raised and Our Response**

Two of the main concerns, the absence of human baselines and the lack of extensive prompt engineering, were already acknowledged as limitations in our original manuscript. We appreciate the reviewers reiterating their importance, which allowed us to expand on them during the rebuttal.
- **Human Baselines**: While explicitly measuring human performance was beyond the initial scope, we clarified how approximate human performance can be inferred (e.g., LLMs surpass humans on CFDQuery but perform far below them on CFDCodeBench and FoamBench). To address this further, we have initiated a human evaluation study and will include its results in the camera-ready version.
- **Prompt Engineering**: We performed additional targeted experiments on the FoamBench Advanced dataset during the rebuttal phase. These showed that prompt refinements led to only marginal improvements (<1.5% absolute), confirming that our original prompts were already effective. The most significant performance gains came from agentic tools (RAG, Reviewer), suggesting that LLM shortcomings stem primarily from reasoning and tool-use rather than prompt design.
- **Metric Choices (NMSE thresholds)**: Reviewers questioned the rationale behind selecting 10% and 30% as NMSE cutoffs. We clarified that these thresholds are well-established in CFD engineering practice, where errors below 10% are typically considered accurate and those above 30% are deemed unacceptable. Moreover, our empirical sensitivity analysis (included in the rebuttal) further validated these choices, demonstrating that they yield stable and interpretable success metrics.

*We hope these clarifications prove to be helpful in recognizing the significance and readiness of our work for acceptance*.

---

### Decision · Program_Chairs · 2025-09-18

**Decision:**

Reject

**Comment:**

This paper introduces CFDLLMBench, an LLM benchmark for computational fluid dynamics (CFD). It consists of three parts: (1) a knowledge part consisting of graduate-level multiple choice questions; (2) simulation problems that must be solved by generating Python code; (3) more complex simulation problems that require implementing a full workflow in OpenFOAM, a CFD software suite. Success is measured using executability, relative error/convergence, and for (3) file/folder structure similarity. The resulting benchmark is used to test a variety of open and closed source models, including ablations with agentic tools (RAG, reviewing).

Reviewers are in agreement that this is a useful benchmark that covers an existing gap in the evaluation of LLMs for science. The benchmark is noted for its complex, domain-specific tasks which have been created by domain experts. The paper is well-written, positions the benchmark clearly relative to existing work, and has detailed descriptions of the methodology and ablations.

Some concerns that were raised concerned the NMSE cutoffs, lack of prompt engineering, limited actionable insights, and a lack of coverage (e.g., limited to OpenFOAM and a small number of tasks overall). The authors successfully addressed several of these points in their rebuttal—for instance, by providing a sensitivity analysis to justify their NMSE thresholds—which satisfied most reviewers.

What remains are two main limitations: The lack of a human baseline, though the authors explain that we can reasonably assume that a CFD graduate student/engineer would be expected to be able to solve all tasks, and the absence of clear mitigation strategies for test set contamination, though the low absolute scores suggest widespread memorization is not currently a concern (and the authors have committed to exploring mitigation strategies in the future). As such, although these limitations are real, I don't believe that they are justification enough for rejection. Crucially, the only reviewer that argued for rejection did so with low confidence (2/5). Hence I agree with the remaining reviewers that this paper, albeit borderline, should be accepted.

===== FINAL UPDATE FROM DB Track PCs ====

The final decision for this paper has been taken by the program chairs after consultation with the SACs. All Senior Area Chairs have ranked papers according to the feedback from the AC during the review process. We decided to leave the original meta-review to reflect the opinion of the AC in light of the initial discussions with reviewers and SAC.